# What is the added value of incorporating pleasure in sexual health interventions? A systematic review and meta-analysis

**Mirela Zaneva**[1], **Anne Philpott**[2]*, **Arushi Singh**[2], **Gerda Larsson**[3], **Lianne Gonsalves**[4]

**1** University of Oxford, Oxford, United Kingdom, **2** The Pleasure Project, United Kingdom and India, **3** The Case for Her, Stockholm, Sweden, **4** UNDP-UNFPA-UNICEF-WHO-World Bank Special Programme of Research, Development and Research Training in Human Reproduction (HRP), Department of Sexual and Reproductive Health and Research, World Health Organization, Geneva, Switzerland

* anne@thepleasureproject.org

**Data Availability Statement:** All relevant data (study characteristics, effect sizes) are within the paper and its Supporting Information files (e.g. tables). Our OSF (https://osf.io/kwz8t/) also

## Abstract

Despite billions of dollars invested into Sexual and Reproductive Health and Rights (SRHR) efforts, the effect of incorporating sexual pleasure, a key driver of why people have sex, in sexual health interventions is currently unclear. We carried out a systematic review and meta-analysis following PRISMA guidelines across 7 databases for relevant articles published between 1 January 2005–1 June, 2020. We included 33 unique interventions in our systematic review. Eight interventions reporting condom use outcomes were meta-analyzed together with a method random effects model. Quality appraisal was carried out through the Cochrane Collaborations' RoB2 tool. This study was pre-registered on Prospero (ID: CRD42020201822). We identified 33 unique interventions (18886 participants at baseline) that incorporate pleasure. All included interventions targeted HIV/STI risk reduction, none occurred in the context of pregnancy prevention or family planning. We find that the majority of interventions targeted populations that authors classified as high-risk. We were able to meta-analyze 8 studies (6634 participants at baseline) reporting condom use as an outcome and found an overall moderate, positive, and significant effect of Cohen's d = 0·37 (95% CI 0·20–0·54, p < 0·001; $I^2$ = 48%; $\tau^2$ = 0·043, $p$ = 0·06). Incorporating sexual pleasure within SRHR interventions can improve sexual health outcomes. Our meta-analysis provides evidence about the positive impact of pleasure-incorporating interventions on condom use which has direct implications for reductions in HIV and STIs. Qualitatively, we find evidence that pleasure can have positive effects across different informational and knowledge-based attitudes as well. Future work is needed to further elucidate the impacts of pleasure within SRHR and across different outcomes and populations. Taking all the available evidence into account, we recommend that agencies responsible for sexual and reproductive health consider incorporating sexual pleasure considerations within their programming.

includes code for effect size computation, as well as screening and search materials. Our pre-registration is available on PROSPERO (ID: CRD42020201822).

**Funding:** The Case for Her (http://www.thecaseforher.com/) provided funding to MZ as an independent consultant in order to carry out a review with high methodological rigor. Funding was not conditional on or tied to results. The funder was not involved in designing the current study, data extraction, analysis, interpretation or write up. One former employee from The Case for Her (GL) served as one of our five independent raters for the abstract and full-text screening stages but was not involved in the final consensus meeting, data extraction, analysis, or interpretation. The funding organization was blinded to analysis and interpretation stages prior to manuscript submission beyond the publicly available information on Prospero. The Case for Her is a Swedish membership organization that gives grants to research and advocacy around women's education and health. The funders had no role in study design, data collection and analysis, decision to publish, or preparation of the manuscript.

**Competing interests:** GL was a salaried employee at the Case for Her, outside the submitted work. MZ reports receiving personal fees as a consultant from the Case for Her during the conduct of the submitted work.

## Background

To date, despite the billions of dollars in domestic and donor funding spent each year to advance SRHR services and programming, and considerable advances in global policy commitments (e.g. through the International Conferences on Population and Development [1], the Commission on the Status of Women, the Millennium Development Goals and subsequently the Sustainable Development Goals), sexual pleasure has been insufficiently addressed. The Guttmacher-Lancet Commission on SRHR in 2018 acknowledged that certain aspects of sexual health, including sexual pleasure were "largely absent from organised SRHR programmes and their links to reproductive health . . . understudied".

An earlier evidence synthesis in 2006 focusing specifically on condom eroticization found that interventions including condom eroticization could lead to improved risk-preventative attitudes. In a meta-analysis of five studies reporting unspecified condom use, condom eroticizing interventions had a positive effect (d = 0·25, 95% CI 0·09–0·42).

## Present study

To our knowledge, this is the first systematic review of interventions incorporating pleasure beyond condom eroticization. Our systematic review and meta-analysis provide evidence for the effectiveness of different interventions incorporating pleasure for a variety of outcomes related to behavior, attitudes, and knowledge in the context of sexual health. Our meta-analysis focuses on condom use as an outcome exclusively and aggregates eight interventions whose control group is either standard care or a matched control group isolating the role of pleasure. Thus, we are able to assess the value added of incorporating pleasure above and beyond standard care interventions. We further provide a quality assessment of the existing evidence base and report on how interventions vary in the degree to which they incorporate pleasure.

## Implications

Interventions that incorporate pleasure have moderate positive and significant effects on condom use and can improve a variety of motivational, informational, and attitude-related outcomes. Future research in the field of SRHR should consider ways to add pleasure in their programming. Further, we find that published, randomized trials of pleasure-incorporating interventions are most prevalent in populations considered at high-risk in some geographical contexts and we point to large gaps in our evidence base, including populations otherwise well-studied in the SRHR literature, such as women of a reproductive age, heterosexual individuals, and members of the general population who are not considered to be at high-risk for HIV/STIs. We recommend that further work consider adopting measures beyond self-report (such as biological markers) and examine dose-response effects for a spectrum of interventions incorporating pleasure. Future SRHR work should consider incorporating pleasure via a range of different interventions and in different populations.

## Introduction

Decades of global commitments, investments, research, advocacy, and innovation–such as the International Conference on Population and Development in Cairo[1] the Fourth World Conference on Women in Beijing [2], the Millennium Development Goals, and most recently, the Sustainable Development Goals [3] and the Generation Equality Forum [4]–have defined modern-day sexual and reproductive health and rights (SRHR), including HIV, programming and services. Despite this, sexual pleasure, a key reason why people have sex, remains insufficiently addressed in most areas of the world [5–8]. This gap has also been acknowledged by

the Guttmacher-Lancet Commission on SRHR in 2018 [9] where aspects of sexual health, including sexual pleasure, were considered as "largely absent from organised SRHR programmes and their links to reproductive health . . . understudied".

The Commission's recent conceptualization of sexual health also included sexual pleasure as a core component [9]. However, education and programming around sexual and reproductive health (including HIV) often defaults to ill-health prevention. For example, programs highlight the dangers and risks of unprotected sex or so-called 'sexually-risky behavior', messages warn of disease burden from HIVs/STIs, and reflect an assumption that sexual decision-making is driven by rational health considerations. By contrast, concerns about sexual pleasure or perceived reductions in libido are often cited as reasons for not using condoms [10] or discontinuing contraception [11].

Assessing the potential value added from incorporating pleasure within SRHR is long overdue but has possibly never been more relevant in the public health discourse. In the months following the start of the ongoing COVID-19 pandemic, many national and subnational health authorities issued guidance on COVID-safe sexual activity. Recommendations addressed multiple types of partners (Netherlands, Ministry of Health, Welfare and Sports [12], Ireland, Sexual Health and Crisis Pregnancy Programme [13]); use of sex toys and cybersex activity (Colombia, Ministry of Health and Social Protection [14]); masturbation [13]; consensual cybersex activity [14]; and non-reproductive sexual practices with a partner that were more (or less) COVID-safe (Spain, Ministry of Health [15]; Argentina, Ministry of Health [16]). In the face of an international public health crisis, these pandemic-issued recommendations by necessity addressed the reality and common questions around pandemic-safe sexual activity, and in doing so acknowledged links between sexual activity and intimate connection, sexual desire, and sexual pleasure.

Exploring the potential of sexual pleasure considerations to make SRHR programming more effective is also particularly timely, considering that fewer than ten years remain to achieve the 2030 target set by the United Nations' Sustainable Development Goals (specifically SDGs 3·7 and 5·6). At present, the most comprehensive evidence available regarding the role of pleasure within SRHR interventions comes from an earlier evidence synthesis [17]. While this review was narrow in focus and assessed the impact of condom eroticization exclusively, it provided preliminary evidence that such forms of pleasure incorporation could have positive effects on certain health-related outcomes. In particular, in meta-analysis of five studies reporting unspecified condom use, condom eroticizing interventions had a significant moderate positive effect (d = 0·25, 95% CI 0·09–0·42). Overall, we consider that at present the effects of incorporating pleasure, including forms of condom eroticization, within SRHR programming are understudied, inconclusive, and warrant further elucidation. As such, our systematic review aims to evaluate the potential impact of incorporating pleasure components within SRHR programming.

## Methods

### Search strategy and selection criteria

We followed PRISMA guidelines [18] in carrying out this systematic review and meta-analysis. We searched 7 databases (PubMed, CINAHL, Sociological Abstracts, PsycINFO, and EMBASE, Global Health, Child Development and Adolescent Studies) for relevant articles published between 1 January 2005–1 June 2020. As a secondary search strategy, we contacted known researchers in the field via email, searched Google Scholar and carried out reference tracking. S1 Appendix reports our search strategy. This was a hybrid approach incorporating both subject headings and keywords. We did not explicitly search for sexual pleasure related

terms as this would potentially exclude interventions that have not been indexed appropriately or do not report sexual pleasure in their abstract, title or keywords. Instead, we opted for an expansive approach where we would first identify sexual health interventions with appropriate design through abstract and then full text screening. Interventions incorporating pleasure would be a subset of all sexual health interventions and would thus be captured by our search. We added an additional full text screening step to specifically identify interventions incorporating pleasure, ensuring robustness and high agreement in the process (see Fig 1). We included SRHR interventions that had test arms with at least one component that incorporated considerations of sexual pleasure. We viewed sexual pleasure in line with the definitions of the World Association for Sexual Health [19] that "sexual pleasure is the physical and/or psychological satisfaction and enjoyment derived from shared or solitary erotic experiences, including thoughts, fantasies, dreams, emotions, and feelings" and the Global Advisory Board for Sexual Health and Wellbeing [20], according to which "sexual pleasure is the physical and/or psychological satisfaction and enjoyment derived from solitary or shared erotic experiences, including thoughts, dreams and autoeroticism".

We included randomized controlled trials and quasi-experimental studies with both pre- and post-intervention measures and a control group published in peer-reviewed journals. For comparison arms, we accepted either no treatment or a non-pleasure incorporating intervention (in SRHR or another area of health) for our systematic review. In order to quantitatively

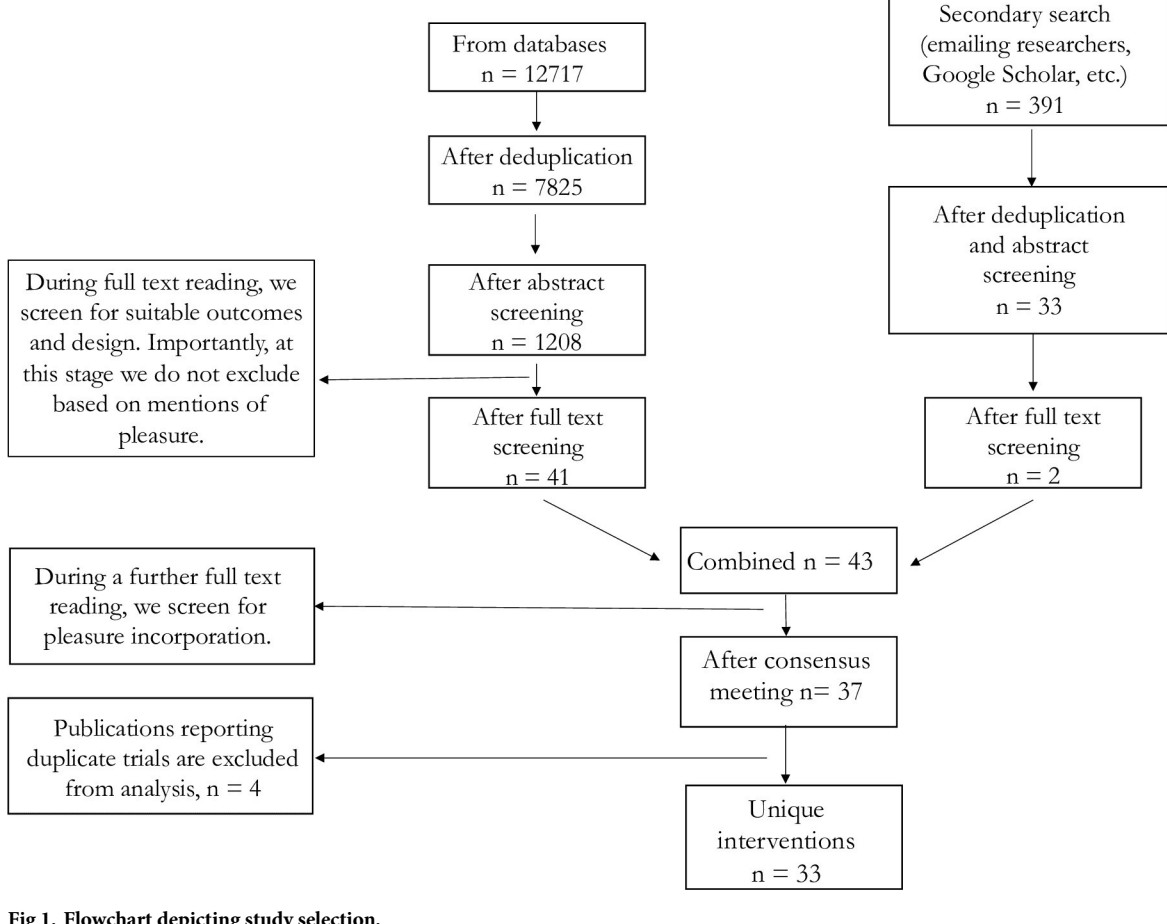

**Fig 1. Flowchart depicting study selection.**

assess the value added from incorporating pleasure, we further restricted the types of eligible comparison arms for the meta-analysis. For this, we accepted only standard SRHR control groups or matched control SRHR groups, as long as they did not include any component(s) with pleasure. For a control group to be considered matched, it had to provide an SRHR intervention similar in duration and implementation form to the test arm. We considered various outcomes including behavioral measures (use of condoms, prevention services, risky behavior, etc.), attitudes and knowledge (about contraception use, STI/HIV incidence, etc.). We did not exclude on the grounds of participant characteristics or language of publication.

The search strategy was carried out by MZ. MZ screened all abstracts and full texts. AP, AS, GL, LG served as independent raters. LG independently screened a random sample of 25% of abstracts and full texts. AP, AS, GL all independently screened a random sample of 10% of abstracts and full texts. Inter-rater reliability was consistently high between MZ and all raters across both stages (ranging from Cohen's Kappa $\kappa = 0\cdot74$ to $\kappa = 0\cdot85$, all $p < 0\cdot001$.)

## Data analysis

We extracted data from the publications (and where needed, intervention manuals and curricula) via a customized spreadsheet recording the pre-registered study characteristics and outcomes of interest. Where data was missing, we contacted authors. Data extraction and double-checking was carried out independently by three reviewers (MZ, AS, LG).

We anticipated considerable heterogeneity amongst studies and planned to carry out an inverse variance random effects meta-analysis (R, version 3.6.1) on the quantitative data for each outcome type for which we could extract an effect size estimate (Cohen's $d$) and 95% confidence interval from a minimum of 3 studies. Study variability was assessed via $I^2$ estimate of heterogeneity. Two raters (MZ, AP) independently used the Cochrane Collaboration's RoB2 tool [21] to examine possible sources of risk of bias. We further narratively synthesized all included studies, with particular consideration for differences in populations, interventions, and ways of incorporating pleasure.

## Results

We screened 7825 abstracts for suitable outcomes and appropriate design and retained 1208 articles for full-text screening. After full text screening, we retained 41 articles and held an additional consensus meeting where all team members had to agree if interventions incorporated pleasure or not. Our supplementary search provided three additional suitable interventions. We found a total of 37 relevant papers (approximately 0·5% of our total screened abstracts) of which 33 reported unique interventions (18886 participants at baseline, see flowchart in Fig 1). Papers reporting duplicate interventions were retained for the purposes of examining the different ways pleasure can be described (see appendix). However, we discarded four duplicate interventions for the systematic review and meta-analysis, retaining only the paper reporting the highest total sample size. S1 Table reports study characteristics. All interventions were implemented in a risk reduction context and targeted HIV/STI-related outcomes. Twenty-five interventions were delivered in the USA with only eight delivered elsewhere: two in South Africa [22, 23], and one in Brazil [24], Spain [25], the United Kingdom [26], Singapore [27], Nigeria [28], Mexico [29].

We were able to meta-analyze eight studies (6634 participants at baseline) that examined condom use as an outcome and had standard care or a matched control group that would allow us to isolate the role of pleasure. Our random effects model (see forest plot in Fig 2) indicated that interventions incorporating pleasure significantly improved condom use compared

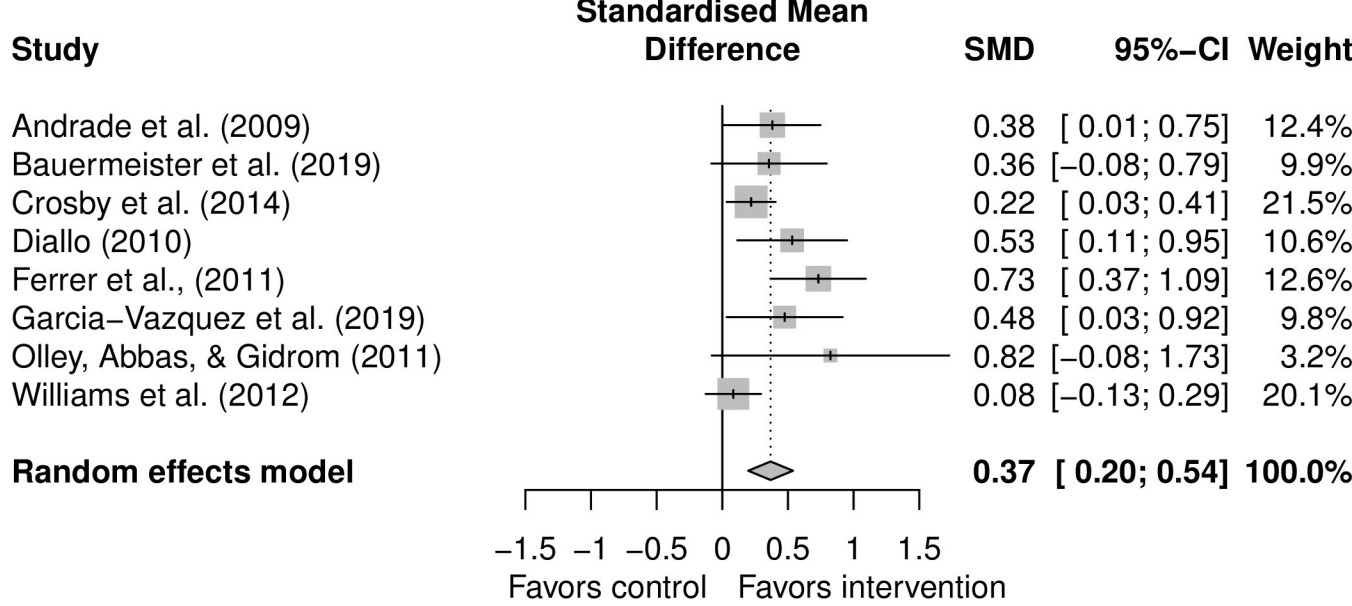

**Fig 2. Forest plot.**

to non-pleasure incorporating interventions (Cohen's d = 0·37, 95% CI 0·20–0·54, p < 0·001, $I^2$ = 48%, $\tau^2$ = 0·043, p = 0·06).

On the basis of the Cochrane Collaboration's RoB2 tool, we found that the methodological quality of the studies varied (see appendix). The most common source of bias was due to outcome measurement (12 out of 33 papers interventions with "Some concerns"), followed by bias from the randomization process (4 interventions with "High risk" and 7 with "Some concerns") and bias from deviations from the intended intervention (11 interventions with "Some concerns"). Nevertheless, the most common assessment for each type of bias was "low risk". Examination of the funnel plot (Figs 3 and 4) suggests that some asymmetry is present and Egger's test supports this (p < 0·05). Duval and Tweedie's trim-and-fill procedure imputes four studies and pooling the effect sizes after this procedure yields an overall effect that remains positive moderate and significant (Cohen's d = 0·22, 95% CI 0·04–0·40, p < 0·05).

We narratively synthesized all studies focusing on differences in populations, types of interventions, and ways of incorporating pleasure. Eight interventions (2633 participants at baseline) exclusively targeted men who have sex with men (MSM). All pleasure-incorporating interventions with MSM were implemented in the USA and a significant body of them focused on young MSM and/or Black or ethnic minority MSM, either living with or without HIV. In general, interventions with MSM achieved successful randomization and had high participant retention rates over multiple follow-up time points. Pleasure was not a focus in these interventions but rather tended to be discussed in the context of behavioral skills such as how condom use could become fun and pleasurable, often considered within a negative and risk-centered context including how alcohol and drugs could affect condom use [30]. A notable exception was the Focus on Future intervention [31] whose primary purpose was to promote condom use in order to enhance sexual pleasure. By and large, these interventions were able to improve some self-reported behavioral outcomes and increases in condom use for anal sex [30–32].

Within the stream of interventions targeting MSM, four online RCT interventions (1298 participants at baseline) targeted HIV prevention. Benefits of online approaches can include the non-judgmental and non-clinical context, as well as the ability to view and interact with

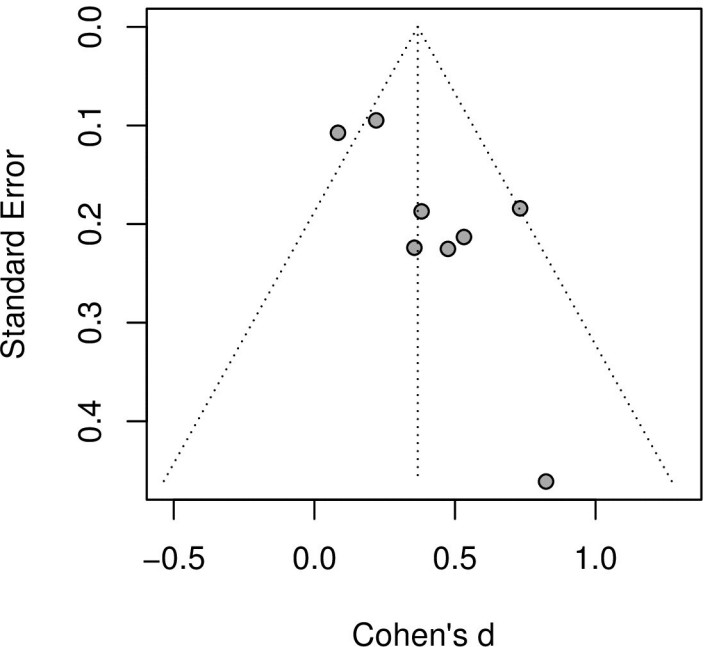

**Fig 3. Funnel plot.**

content at one's leisure. These interventions, MINTS-II [33], HINTS [34], Guy2Guy [35], myDEx [36], provided informational and motivational content and typically discussed pleasure in the context of behavioral skills (e.g., lubrication, safer sex practices). Participants in myDEx's interventional arm were less likely to have engaged in condomless receptive anal sex than controls, and intervention participants in MINTS-II also reported a marginally significant decrease in the number of men with whom they engaged in unprotected anal intercourse, while Guy2Guy did not find behavioral differences between the arms. For sexually inexperienced youth in the Guy2Guy trial, motivation increased compared to controls which may highlight the importance of online interventions particularly for younger and sexually inexperienced youth.

Nine interventions (7473 participants at baseline, delivered in the USA) focused on racial and ethnic minorities rather than sexual partner preferences or sexual behaviors. One intervention was provided to adolescent women from any minority ethnic background who have suffered abuse; the remaining eight interventions focused on African American participants. Within this general categorization, participants included men newly diagnosed with STIs [37], young males attending STI clinics [38], young people and adolescents [39–41], heterosexual men and women [42], and women [43], including women in primary care settings [44]. Beyond the considerable heterogeneity in participant population, intervention differed significantly in terms of duration (e.g., brief or single interventions [37, 38, 40, 43] vs multi-session [30, 45, 46]), and context (e.g., clinical [37, 44] or not [40–43]). Pleasure was incorporated in a range of ways, including addressing negative beliefs about condom use and pleasure, discussing ways to make condom use more pleasurable, and eroticization of safe sex. The noted methodological heterogeneity and variability in target population characteristics poses a challenge for extrapolating an overall trend or conclusions about the effectiveness of this stream of research.

A further five RCT interventions [29, 44, 47–49] (1988 participants at baseline) targeted people who use or have used drugs. These participants often faced multidimensional and intersectional risks due to their ethnicity, work (sex workers [29]), health status (living with HIV

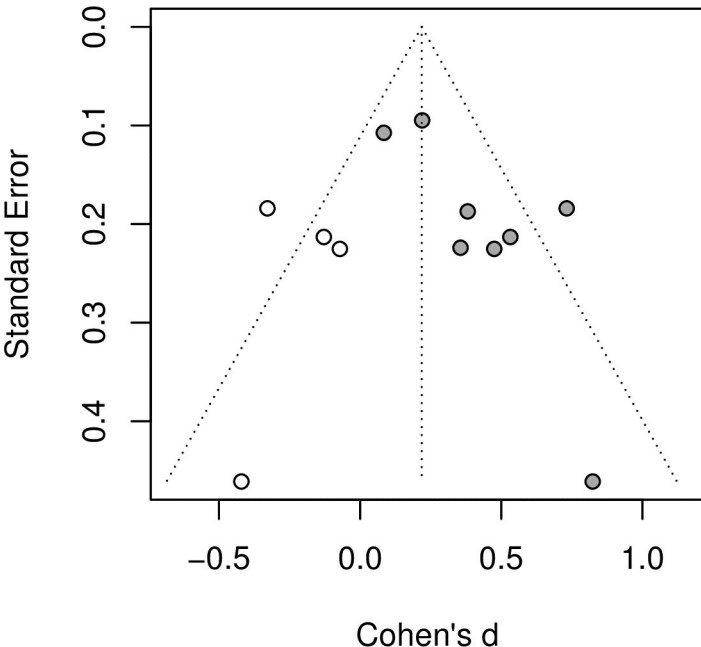

**Fig 4. Funnel plot following trim and fill procedure.**

[50]), socioeconomic status [49], or were otherwise classified as high risk and already enrolled in treatment [47]. Pleasure was often discussed in sessions regarding negotiating harm reduction and how safer sexual practices can be eroticized [29, 47, 50] but could also include a more empowering and rights-based approach such as assessing personal sexual rights, positive sexual choices [50], increasing confidence and pride in one's body, affirming one's right to pleasure [48]. Despite working with high-risk groups who often faced additional stigma and adversities, these interventions were effective in increasing informational outcomes such as knowledge about HIV/STI and self-efficacy. Behavioral outcomes were not assessed in all interventions [48] and were likely affected by response bias [47], although notably amongst participants in a pleasure-inclusive interactive version of one intervention [29] there was a 50% decrease in HIV/STI incidence compared to a control didactic sex intervention. Pleasure-inclusive interventions can have positive effects for high-risk populations, but further empowerment and appropriate interactive interventional programming may be required to adequately support individuals at risk.

Finally, several programs targeted young people and adolescents in schools or educational contexts. Notably, two large-scale quasi-experimental evaluations (3393 participants at baseline) of sexual education programs in Spain [25] and Brazil [24] presented sexuality as a positive human value and source of pleasure and included various forms of empowering and interactive activities and were both able to increase condom use in intervention schools compared to controls. In another school-based RCT [26], teenagers were provided with a condom use promotion leaflet. This brief 20-minute intervention found positive effects in a variety of cognitive domains, such as self-reported attitudes towards condom use, self-efficacy, and intention to use condoms. Changes in behavioral outcomes were not observed at a one-month follow-up, likely due to the short and informational nature of the intervention. Positive intervention effects on cognitive and some behavioral domains were also found in brief educational curricula [40, 41]. In interventions targeting adolescents, populations considered at higher risk by the authors were still prevalent, including ethnic minority girls who have suffered sexual

abuse [45], adolescent girls attending Planned Parenthood [51], and incarcerated youth [52]. While this latter stream of interventions generally tended to have positive effects on informational, motivation, and behavioral outcomes (such as fewer infections), they were also marked by attrition problems. Limited evidence for university students is available from three interventions [22, 53, 54]. Condom promotion video materials that discussed how condom use could be erotic [53, 54] improved condom use in one intervention [55] and improved self-efficacy but had no effect on consistent condom use in another [53]. Further, an eight-session module targeting HIV risk reduction had significant positive effects on frequency of condom use, self-efficacy, and HIV-related knowledge [22].

## Discussion

We provide qualitative and quantitative evidence that SRHR interventions incorporating pleasure can have positive effects on a variety of behavioral and information outcomes. Our meta-analysis indicates that interventions incorporating pleasure have a moderate positive and significant effect on condom use (Cohen's d = 0·37, 95% CI 0·20–0·54, p < 0.001) above and beyond standard care. Affirming human sexuality and the reasons why people have sex could be an important way to ensure sexual health interventions are effective.

Notably, we found large heterogeneity in terms of intervention designs, participant risk profile, as well as time and emphasis placed on pleasure. Hence, it is difficult to extrapolate a conclusive causal mechanism through which pleasure can affect the different health-related outcomes, pertaining to domains such as behavior, attitudes, knowledge, motivation. Relatedly, due to heterogeneity, it is also difficult to isolate enough comparable studies to allow a robust estimation of an effect size. While we have managed to do so for condom use, future work is required in order to better understand the impacts of adding pleasure components to SRHR interventions on further outcomes.

In our narrative synthesis, we find some qualitative evidence that significant positive effects are found in relation to how pleasure was positioned in the interventions. Specifically, the majority of included interventions discuss pleasure in the context of behavioral skills (e.g., making condom use fun or sexy, using lubrication to enhance sexual pleasure) and here, it seems, behavioral outcome changes were most prevalent. Notably, one intervention [26] provided informational content only and found improvements in cognitive and informational domains. Importantly, most research captured under our systematic review focused on HIV and STI reduction. We find an evidence gap for interventions targeting pregnancy prevention and contraception interventions that warrants future attention.

A further consideration in better understanding the effects of pleasure pertains to examining participant populations. We found two educational programs delivered in schools to general population students who were not considered at risk but, due to their age, could be considered vulnerable. All remaining programming targeting adolescents was, by and large, targeted towards adolescents classified as 'at risk' or having high incidence of STIs. Currently, there is an evidence gap for the impacts of interventions incorporating pleasure on the level of the general population, including heterosexual individuals and couples, with a particularly pronounced gap for women of reproductive age as well as older women. Overall, we find that implementing pleasure within SRHR interventions occurs in a largely risk-reduction context with a strong targeting of groups deemed 'high-risk'. We provide further detail on the spectrum of pleasure, examining considerations such as the overall aims of interventions, how much time is dedicated to pleasure, and in what contexts, in S2 Appendix.

In terms of limitations, it is possible that we may have missed interventions incorporating pleasure because their write-ups did not indicate pleasure was a part of the intervention's

components or delivery. The way interventions are described is often guided by considerations linked to funding, publication, cultural and other biases. We acknowledge that such contextual and financial factors may also limit which interventions can be implemented in the first place. We discuss these issues as they relate to challenges for carrying out a systematic review and understanding effectiveness elsewhere [55]. In so far as how intervention description may affect inclusion, we have tried to mitigate the potential impact of this as much as possible by contacting authors and reading intervention manuals. Equally, we anticipate that some interventions might have been delivered in ways that included pleasure beyond what can be captured in intervention manuals. As a further limitation, we are aware of work involving pleasure not published in peer-reviewed journals, as well as interventions whose designs are not eligible for inclusion per our criteria (e.g., cross-sectional or non-controlled work). There are already advocacy and civil society-led efforts to document and connect organizations leading in this type of programming [56]. For this review, we instead chose to prioritize a conservative approach with strict cut-offs in study design in order to maximize the chance of isolating the value-added of incorporating pleasure components as compared with SRHR interventions which do not include pleasure considerations. Notably, in this review, we also examined pleasure through a focused definition, excluding other intervention framings, such as those grounded in a gender transformative, or personal and/or economic empowerment approaches. This should not be perceived as nonrecognition of their importance: good SRHR should include the potential for respectful, equitable, consensual, as well as pleasurable interactions.

Based on the current findings, we are able to point to areas where future work is needed. As a first step, clear description of work that incorporates pleasure, such as through key words and direct in-text descriptions, could facilitate accurate identification and later systematic assessments. Precise definitions of how much emphasis and time have been allocated to pleasure components will additionally allow for a consideration of dose-response effects. Moreover, addressing the existing gaps—in populations (e.g., heterosexual individuals, women), as well as types of programming (i.e., family planning, pregnancy prevention)—is another direct implication. Importantly, considering that the majority of identified outcomes are self-reported, future research could also provide value by considering different measures of impact, including biological markers such as STI incidence.

The current review is also timely in the context of the United Nations' Sustainable Development Goals, particularly goals 3·7 and 5·6, which target universal access to sexual and reproductive health and reproductive rights. Continued high levels of mortality and morbidity attributed to SRH-related outcomes indicate that revisiting how to most effectively design SRH programs and education is warranted. This may involve a fundamental rethink of how programs are oriented. Our review indicates that programs and education which better capture a full working understanding of sexual health, which acknowledges that sexual experiences can be 'pleasurable', have been demonstrated to improve not only knowledge and attitudes around sexual health, but also safer sex practices. With fewer than ten years to go, and many countries not on track to meeting these SRHR goals [57], interventions that incorporate pleasure may prove an important strategy to ensure that not only positive outcomes are obtained, but that they go beyond the effects normally anticipated by standard care programming. Continuing to avoid pleasure inclusive sexual health and education risks further misdirecting or inefficiently utilizing the much-needed resources to reach the SDGs.

## Supporting information

**S1 Table. Study characteristics.**
(DOCX)

**S1 Appendix. Search strategy as implemented in PubMed.**
(DOCX)

**S2 Appendix. Assessment of the different ways pleasure is described in articles.**
(DOCX)

**S3 Appendix. Quality assessments based on the Cochrane Collaboration's RoB2 tool.**
(DOCX)

**S1 Checklist. PRISMA 2009 checklist.**
(DOC)

## Author Contributions

**Conceptualization:** Anne Philpott, Arushi Singh, Gerda Larsson, Lianne Gonsalves.

**Data curation:** Mirela Zaneva.

**Formal analysis:** Mirela Zaneva, Anne Philpott, Arushi Singh, Lianne Gonsalves.

**Methodology:** Mirela Zaneva, Anne Philpott, Arushi Singh, Gerda Larsson, Lianne Gonsalves.

**Project administration:** Mirela Zaneva.

**Resources:** Anne Philpott, Arushi Singh, Gerda Larsson, Lianne Gonsalves.

**Supervision:** Lianne Gonsalves.

**Validation:** Mirela Zaneva, Anne Philpott, Arushi Singh, Lianne Gonsalves.

**Writing – original draft:** Mirela Zaneva.

**Writing – review & editing:** Mirela Zaneva, Anne Philpott, Arushi Singh, Lianne Gonsalves.

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
