## [Decision Letter · Decision Letter 0]

7 Oct 2021

PONE-D-21-23113What is the added value of incorporating pleasure in sexual health interventions? A systematic review and meta-analysis.PLOS ONE

Dear Dr. Philpott,

Thank you for submitting your manuscript to PLOS ONE. After careful consideration, we feel that it has merit but does not fully meet PLOS ONE’s publication criteria as it currently stands. Therefore, we invite you to submit a revised version of the manuscript that addresses the points raised during the review process.

We look forward to receiving your revised manuscript.

Kind regards,

Lucinda Shen, 

Staff Editor

on behalf of 

Limin Mao, PhD

Academic Editor

PLOS ONE

Journal Requirements:

Additional Editor Comments (if provided):

In view of two very positive reports, I command your efforts and suggest the consideration of the feedback below in your final paper.

Reviewers' comments:

Reviewer's Responses to Questions

**Comments to the Author**

1. Is the manuscript technically sound, and do the data support the conclusions?

Reviewer #1: Yes

Reviewer #2: Yes

2. Has the statistical analysis been performed appropriately and rigorously? 

Reviewer #1: Yes

Reviewer #2: Yes

3. Have the authors made all data underlying the findings in their manuscript fully available?

Reviewer #1: Yes

Reviewer #2: Yes

4. Is the manuscript presented in an intelligible fashion and written in standard English?

Reviewer #1: Yes

Reviewer #2: Yes

5. Review Comments to the Author

Reviewer #1: Strong/sound rationale and implications for the study. Well written. Nicely links with UN goals/larger conversations re health/sexual health. Good methodological approach re review. Good rigour with inclusion of meta analysis. Findings coalesce with focus on enhanced sex-positive messaging and inclusion of intimacy/pleasure etc as other relevant needs associated with sexual practices. Narrative synthesis of main findings from papers included in the review is well written. Discussion is sound.

Reviewer #2: Thank you for the opportunity to review this excellent paper. It is an interesting and important topic, concisely and clearly written. It uses sound methodology and results are explained clearly with an appropriate level of detail included.

I have three very minor suggestions, which I would consider optional to revise:

1. One area could be added to in the Discussion – the lack of inclusion of pleasure in sexuality education is probably not due to its effectiveness, but more about political and moral judgements. This could be discussed further.

2. This might be an issue with the format of the proof, but I can’t see any titles/legends on the figures.

3. It would be helpful if the 8 studies included in the meta-analysis were identified in the table somehow so they can be easily found.

6. PLOS authors have the option to publish the peer review history of their article (what does this mean?). If published, this will include your full peer review and any attached files.

Reviewer #1: **Yes: **Associate Professor Amy B. Mullens

Reviewer #2: **Yes: **Megan Lim

---

## [Author Response · Author response to Decision Letter 0]

18 Nov 2021

thank you for the helpful comments - we have addressed them and submitted a letter with responses to reviewers.

---

## [Editor Report · Decision Letter 1]

23 Nov 2021

What is the added value of incorporating pleasure in sexual health interventions? A systematic review and meta-analysis.

PONE-D-21-23113R1

Dear Dr. Philpott,

We’re pleased to inform you that your manuscript has been judged scientifically suitable for publication and will be formally accepted for publication once it meets all outstanding technical requirements.

Kind regards,

Limin Mao, PhD

Academic Editor

PLOS ONE
---

## [Editor Report · Acceptance letter]

6 Jan 2022

PONE-D-21-23113R1 

What is the added value of incorporating pleasure in sexual health interventions? A systematic review and meta-analysis. 

Dear Dr. Philpott:

I'm pleased to inform you that your manuscript has been deemed suitable for publication in PLOS ONE. Congratulations! Your manuscript is now with our production department. 

Kind regards, 

on behalf of

Dr. Limin Mao 

Section Editor

PLOS ONE